# Melatonin versus Sleep Deprivation for Sleep Induction in Nap Electroencephalography: Protocol for a Prospective Randomized Crossover Trial in Children and Young Adults with Epilepsy

**DOI:** 10.3390/metabo13030383

**Published:** 2023-03-04

**Authors:** Costanza Varesio, Valentina Franco, Ludovica Pasca, Massimiliano Celario, Cinzia Fattore, Guido Fedele, Paola Rota, Michela Palmisani, Valentina De Giorgis

**Affiliations:** 1Department of Child Neurology and Psychiatry, IRCCS Mondino Foundation, 27100 Pavia, Italy; 2Department of Brain and Behavioral Sciences, University of Pavia, 27100 Pavia, Italy; 3Clinical and Experimental Pharmacology Unit, Department of Internal Medicine and Therapeutics, University of Pavia, 27100 Pavia, Italy; 4IRCCS Mondino Foundation, 27100 Pavia, Italy; 5AFI—Associazione Farmaceutici dell’Industria (AFI), 20100 Milan, Italy; 6Department of Biomedical, Surgical and Dental Sciences, University of Milan, 20100 Milan, Italy; 7Institute for Molecular and Translational Cardiology (IMTC), San Donato Milanese, 20097 Milan, Italy

**Keywords:** melatonin, 6-hydroxymelatonin, sleep latency, nap electroencephalography, epilepsy, hypno-induction

## Abstract

Electroencephalography (EEG) continues to be a pivotal investigation in children with epilepsy, providing diagnostic evidence and supporting syndromic classification. In the pediatric population, electroencephalographic recordings are frequently performed during sleep, since this procedure reduces the number of artifacts and activates epileptiform abnormalities. To date, no shared guidelines are available for sleep induction in EEG. Among the interventions used in the clinical setting, melatonin and sleep deprivation represent the most used methods. The main purpose of this study is to test the non-inferiority of 3–5 mg melatonin versus sleep deprivation in achieving sleep in nap electroencephalography in children and young adult patients with epilepsy. To test non-inferiority, a randomized crossover trial is proposed where 30 patients will be randomized to receive 3–5 mg melatonin or sleep deprivation. Each enrolled subject will perform EEG recordings during sleep in the early afternoon for a total of 60 EEGs. In the melatonin group, the study drug will be administered a single oral dose 30 min prior to the EEG recording. In the sleep deprivation group, parents will be required to subject the child to sleep deprivation the night before registration. Urinary and salivary concentrations of melatonin and of its main metabolite 6-hydroxymelatonin will be determined by using a validated LC-MS method. The present protocol aims to offer a standardized protocol for sleep induction to be applied to EEG recordings in those of pediatric age. In addition, melatonin metabolism and elimination will be characterized and its potential interference in interictal abnormalities will be assessed.

## 1. Introduction

Neurophysiological investigations in children usually require being performed during sleep to minimize the need for patient cooperation [1,2]. This is particularly true when electroencephalography (EEG) is concerned. EEG should ideally be performed in both awake and sleep states. The EEG recording during sleep is relevant as it is useful to (i) reduce the number of artifacts due to a relative lack of cooperation; (ii) maximize the diagnostic yield of the test since it acts as an activating procedure unraveling epileptic abnormalities; and (iii) reveal many childhood epileptic syndromes present with peculiar characteristics during sleep [1,2,3].

As routine nap EEGs are mainly performed during daytime hours, sleep is usually achieved either by sleep deprivation or pharmacological agents [4,5]. Sleep deprivation is usually partial, but it can be extremely burdensome for patients and families, especially in children with neurodevelopmental disorders or behavioral problems [6]. Pharmacological agents have been mainly represented by barbiturates, chlorpromazine, and chloral hydrate [7]. However, these agents interfere with sleep macrostructure and might affect the interpretation of EEG recordings by distorting background activity or by suppressing epileptiform activities. Moreover, they might induce persistent sleepiness. To overcome these inconveniences, during the last two decades, melatonin has gained progressive interest for nap EEG recordings [8].

Melatonin (5-methoxy-N-acetyltryptamine) is a pineal hormone that regulates circadian rhythms and the sleep–wake cycle, and is widely used to treat sleep disturbances. A melatonin sleep-inducing effect for nap EEG procedures has been reported in adults and children, together with substantial safety and tolerability [9,10,11]. Moreover, it has been reported not to affect the occurrence of epileptiform abnormalities [12].

In a quite recent survey aiming at describing the strategies adopted in everyday clinical practice for sleep induction in different pediatric and adult epilepsy centers in Italy, it emerged that melatonin is the most commonly used drug for sleep induction [5]. However, there is no consensus on the timing and doses of melatonin to be administered for EEG recordings. Doses of melatonin vary substantially according to different studies, ranging from 2 mg up to 10 mg [8].

In this context, guidelines for melatonin use and dosage in the specific population of pediatric patients undergoing sleep EEG recordings are claimed.

This randomized study aims to assess the efficacy and safety of melatonin against sleep deprivation in inducing sleep for EEG recordings in a cohort of children and young adult outpatients with epilepsy in a tertiary-center pediatric epilepsy unit and to provide recommendations for the regular use of melatonin for nap EEG recordings. In addition, melatonin interference in seizure manifestations and interictal abnormalities will be assessed.

## 2. Methods

### 2.1. Study Design

The proposed study is a monocentric randomized crossover non-inferiority trial. Patients will be randomly allocated in a 1:1 ratio to start the study with melatonin or sleep deprivation for hypno-induction to perform EEG recordings. Subjects who will be allocated to the first sequence (sequence 1, *n* = 15) will receive melatonin in period 1 and sleep deprivation in period 2, while subjects who will be allocated to the second sequence (sequence 2, *n* = 15) will receive the two interventions in the reverse order. A total of 60 EEGs will be obtained.

The Consolidated Standards of Reporting Trials (CONSORT) flow diagram is shown in Figure 1 [13].

### 2.2. Study Setting

Subjects will be enrolled from the Child Neurology and Psychiatry Unit of the IRCCS Mondino Foundation (Pavia, Italy). The study was approved by the local Ethics Committee (Reference N°: P-20200099096) and registered at https://clinicaltrials.gov (accessed on 15 January 2023) (NCT05654415). All of the researchers involved in the study are trained in study procedures and Good Clinical Practice and Good Laboratory Practice guidelines. The clinical team has consolidated experience in the management of patients with epilepsy and will be responsible for outcome evaluation. All of the trial procedures that will be applied have been validated. Study data will be collected in case report forms (CRFs) designed ad hoc and entered into a dedicated database.

### 2.3. Participants

Subjects will be screened by using the following inclusion criteria: (i) children and young adults with epilepsy aged between 4 and 18 years with normal psychomotor development; (ii) stable seizure frequency in the 3 months preceding the study; (iii) stable antiseizure medication (ASM) treatment in the 3 months preceding the study; and (iv) written informed consent from parents or the legal representative.

Exclusion criteria are as follows: (i) subjects diagnosed with obstructive sleep apnea or sleep disturbances; (ii) history of neurodevelopmental disorders; (iii) concomitant use of hypnotics, stimulants, systemic corticosteroids, or other immunosuppressants; (iv) history of daily melatonin use; (v) any condition which, in the opinion of the investigator, would compromise the achievement of the objectives of the study; and (vi) failure to obtain the informed consent.

### 2.4. Interventions

Melatonin will be compared to sleep deprivation. Patients who fulfill the inclusion criteria will be randomly allocated to receive melatonin or sleep deprivation as the first intervention. One–three months after the first EEG recording, the crossover will take place. Melatonin oral solution at the dose of 3 or 5 mg depending on body weight (3 mg for patients < 15 kg; 5 mg for patients > 15 kg) will be used as the study medication. Melatonin will be administered as a single dose 30 min before the EEG recording. Sleep deprivation will be represented by sleep deprivation of 50% of physiological sleep the night before EEG registration. The investigator a week before the EEG test will contact the patient’s caregiver. The telephone call will be devoted to investigating the patient’s sleep habits in terms of mean amount of sleep per night, excluding the presence of concomitant sleep disturbances. Based on the reported mean sleep time per night, caregivers will be instructed to deprive 50% of sleep the night before the EEG. Moreover, caregivers will be asked to not let patients perform naps in the morning of the examination or on their way to the hospital. Each EEG recording will be performed at the same hour, in the early afternoon (1:30 p.m.). The EEG recording will last 1 h. All EEG recordings will be performed in a dark and quiet room, and patients will be instructed to lie calmly and try to sleep. EEG electrodes will be placed on the patient’s scalp surface according to the 10/20 system. Polygraphic channels including ECG and respiration will be recorded for all participants. Electromyogram activity will be recorded according to the patient’s epileptic syndrome.

### 2.5. Primary Outcome Measures

To determine the non-inferiority of 3–5 mg melatonin solution compared to sleep deprivation alone in sleep induction and maintenance, the primary outcome measure will be sleep onset latency (defined as the time in seconds elapsing from full weakness to the earliest signs of non-REM stage 2 sleep as recorded by the EEG) [14].

### 2.6. Secondary Outcome Measures

Secondary outcomes will be as follows:-Wake after sleep onset (time in minutes patients are awake after sleep onset, WASO).-Sleep duration (duration in minutes from the beginning of non-REM stage 2 sleeping to the time patients are roused at the end of the study or spontaneously wake).-Number of EEGs characterized by electroclinical seizures collected and classified according to the 2017 International League Against Epilepsy (ILAE) classification and terminology [15].-Interictal abnormalities defined as epileptiform spike density per hour.

### 2.7. Melatonin and 6-Hydroxymelatonin Determination via Ultraperformance Liquid Chromatography Coupled to Mass Spectrometry

Urinary and salivary concentrations of melatonin and of its main metabolite 6-hydroxymelatonin will be measured in samples collected from all patients by using a validated LC-MS method with minor modifications [16].

Urinary and unstimulated oral fluid samples will be taken 30 min after the EEG. Urine samples will be collected in a 50 mL polypropylene tube. Samples will be centrifuged at 17,000× *g* for 10 min and the supernatants will be transferred in 10 mL tubes. Oral fluid sampling will be obtained via passive drooling, aspirated with a syringe, and transferred into 2 mL polypropylene tubes. Subjects will be instructed to not eat/drink for 30 min before sample collection and to rinse their mouths with plain water. All samples will be stored at −80 °C until analysis.

### 2.8. Sample Size Calculation

The sample size was calculated based on the primary efficacy endpoint considering the crossover design and the non-inferiority hypothesis. A recent study demonstrated that the combination of sleep deprivation and melatonin was more effective compared to each method alone for inducing sleep [17]. In a previous study, Jain and collaborators demonstrated that melatonin was associated with improvements in sleep latency and wakefulness after sleep onset [18]. In particular, in the latter study, melatonin reduced sleep latency by about 11 min. In both studies, the corresponding calculated 95% confidence lower limit was about −3 min. Given this effect in the two aforementioned studies [17,18] and considering preliminary data at our clinical center, we assume that the clinical limit for no relevant differences in efficacy between the study procedures is −3 min in the sleep latency onset. The sample size was calculated using the formula for non-inferiority crossover studies reported by Julious [19]. A sample of 60 EEGs (30 for each study sequence) to achieve a power of 80% and a level of significance for a one-sided test of 0.025 will be sufficient to demonstrate the non-inferiority of the group treated with melatonin compared with the group of participants randomized to receive sleep deprivation [19].

### 2.9. Randomization

Subjects will be randomly assigned to melatonin or sleep deprivation for the first recording using R software (R Core Team 2021; R Foundation for Statistical Computing, Vienna, Austria) whereby random numbers will be generated. A computerized algorithm will be implemented taking into account the two sequences of the crossover design. A casual assignment to one of the two sequences (sleep deprivation–melatonin and melatonin–sleep deprivation) will be applied without any correction for demographic or clinical factors.

### 2.10. Ethical Considerations

Written informed consent will be obtained from subjects/parents of all subjects. Child neurologists and neurophysiologists participating in the investigation will recruit participants and collect informed consent. Parents will be fully informed about the study during a counseling session, and an information sheet will be provided.

### 2.11. Statistical Methods

Data will be analyzed initially using descriptive statistics. Dichotomous data will be compared using the chi-square test and parametric or non-parametric inferential tests for continuous data will be used after evaluation on normal distribution/homogeneity of variance. The method proposed by Grizzle will be applied for testing treatment differences [20]. Considering that the study was designed as a 2 × 2 crossover, potential carry-over effects will be considered in the inferential model used for the analysis of the primary endpoint. A *p*-value ≤ 0.05 will be considered to be statistically significant. A 95% confidence interval for non-inferiority will be appropriately calculated. All analyses will be performed using SAS 9.4 software (SAS Institute Inc., Cary, NC, USA).

## 3. Discussion

Sleep EEG recording is an essential diagnostic tool in outpatient pediatric epilepsies to increase the diagnostic yield of the procedure [3]. Many epileptic syndromes with onset in childhood and adolescence present epileptiform abnormalities which are activated or appear to be more prominent during sleep [1,2,3]. Moreover, obtaining sleep allows for improving the technical quality of the recording and minimizing artifacts, especially in those patients for whom it is difficult to obtain an adequate level of collaboration, as in the case of children and/or patients with neurodevelopmental or behavioral disorders [3,4]. In routine clinical practice, outpatient EEGs, including sleep EEGs, are performed during daytime, thus implying the need to perform sleep EEGs in the form of nap EEGs [2]. In this circumstance, although spontaneous sleep might be more desirable, it might be difficult for patients to achieve sleep spontaneously.

Despite the necessity of attaining sleep during EEG recordings, there are to date no guidelines or recommendations addressing strategies for sleep induction [5]. The clinical practice among neurophysiology centers worldwide is substantially disparate both in terms of methods and applications [8]. Studies conducted in the field are likewise extremely heterogeneous, making it utterly difficult to draw evidence-supporting strategies for sleep induction in subjects undergoing EEG procedures due to their observational and/or retrospective design or lack of randomization and inhomogeneous population and protocols. Hence, more rigorous studies are needed to gather conclusions and inform clinical practice in the field. In a quite recent survey conducted in neurophysiology centers in Italy [5], it emerged that the most common strategies for sleep induction are represented by sleep deprivation and melatonin administration. In this context, our study protocol was designed to compare these two methods in a specific subset of pediatric epileptic outpatients. Sleep deprivation is usually applied as partial deprivation [2,21], as it is more tolerable and feasible for pediatric patients and their families as compared to total sleep deprivation. However, partial sleep deprivation also might be extremely burdensome for patients, especially of younger-age children and those with neurodevelopmental and behavioral disorders, and their families. Sleep deprivation might also increase the risk of eliciting seizures [6]. In addition, compliance with the indications provided could be suboptimal due to the difficulties encountered by parents in keeping the children awake the night before the recording. Melatonin has gained attention in the last two decades as a possible alternative to sleep deprivation in EEG laboratories due to its favorable efficacy and safety profile. Melatonin is a pineal hormone with various properties including the regulation of circadian rhythms and the sleep–wake rhythm [9]. Melatonin has been demonstrated to be safe and short treatment at low doses has almost no side effects in the healthy population [10]. It does not alter sleep macrostructure and background activity significantly [22]. A diagnostic yield of the sleep EEG and the presence of paroxysmal epileptiform abnormalities has been reported not to be modified by melatonin administration [12,22,23,24]. To date, few studies have been devoted to comparing the efficacy of partial sleep deprivation and melatonin in sleep induction in the pediatric EEG setting. These studies showed inconsistency due to small sample sizes and heterogeneity in study designs: observational case–control study with unselected assignment [22,23], retrospective comparison [24], prospective cohort study [17,25] and prospective crossover study [12]. Additionally, the populations included in the studies are quite heterogeneous, varying from patients with epilepsy or unclear paroxysmal events [24] to all patients admitted for sleep EEG [12,17,22,23,25]. Our choice is to restrict the study population to pediatric patients with epilepsy, a stable clinical and pharmacological picture, and without cognitive, neurodevelopmental, or sleep disorder comorbidities to reduce possible confounding factors.

Doses of melatonin were administered at different dosages depending on age in most of the studies, varying between 2 mg and 10 mg [12,17,22,23,24,25].

There are no studies on the potential dose-dependent effects of melatonin during acute use as a hypnotic in children. Based on our clinical experience, we will administer melatonin according to body weight (3 mg for patients < 15 kg; 5 mg for patients > 15 kg). Our choice to not exceed 5 mg is supported by the observation by Eiserman and colleagues [25] that the administration of higher doses of melatonin showed comparable success rates in hypno-induction with regard to lower doses [22,23,24]. The timing of melatonin administration reported in different studies is substantially variable, ranging from 30–40 min before EEG recording [12], 15 min before electrode application [24], or at the time of electrode application [22,23]. We will provide melatonin 30 min before EEG execution because when exogenous melatonin is administered, it reaches peak plasma concentration in 20–120 min, and its sleep onset effect appears 30–35 min after drug administration [11]. Additionally, the extent of sleep deprivation is extremely variable, ranging from a few hours to almost 10 h. Even within single studies, the timing and extent of sleep deprivation might be inconstant [23]. The approximate need for sleep varies significantly from childhood to adolescence [26]. It is estimated that in preschool children, nocturnal sleep time begins to average at 10 h, whereas as children grow up, nocturnal sleep time decreases, becoming 8 h per night in adolescence. A variety of factors influence sleep duration in children, adolescents, and young adults, including generational trends, societal pressures (such as school start times), and familial habits [27]. Taking this huge variability in mind and considering the wide age range of enrolled patients, our choice is not to establish rigid times for going to bed and waking up before the EEG, but to provide a more flexible indication, represented by the deprivation of 50% of physiological sleep for each single patient. Additionally, nap EEGs are usually performed at variable times of the day [23,24] or the time of execution is not specified [17]. Although there are no clear data on the best time of day to perform nap EEGs, in the context of our protocol, we have decided to perform all EEG recordings in the noon session to exploit the “afternoon nap zone” of the physiological circadian rhythm [28]. Although melatonin appeared to be non-inferior compared with sleep deprivation in previous studies, variability in melatonin dosing and timing and differences in the application of sleep deprivation and subject selection, as well as other factors such as environmental differences and the study population, have so far prevented the drawing of a solid conclusion and the provision of a hypno-induction protocol shared among the different pediatric neurophysiology centers. Our study was developed according to a randomized crossover non-inferiority design to provide evidence for a standardized protocol to be used in the routine clinical setting for sleep induction in nap electroencephalography. Furthermore, the thorough estimate of the main metabolite 6-hydroxymelatonin will allow us to further characterize melatonin metabolism and elimination, providing useful information on the optimal dose and time of administration. The findings of the proposed trial will ultimately establish melatonin interference in interictal abnormalities during acute use as a hypnotic agent.

## Figures and Tables

**Figure 1 metabolites-13-00383-f001:**
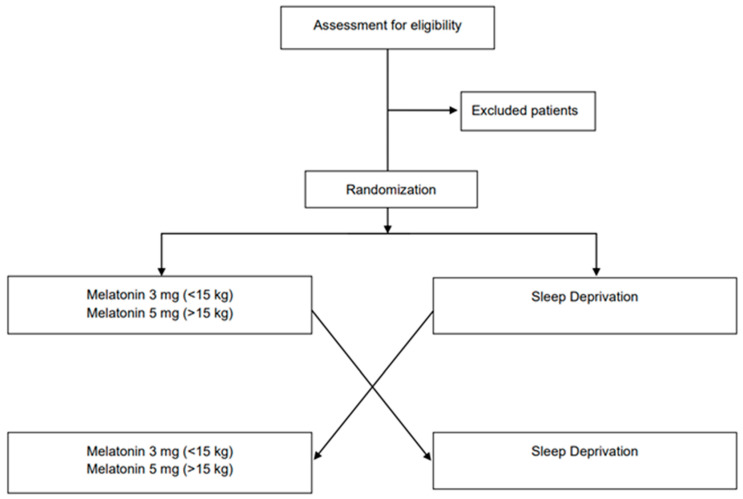
CONSORT flowchart for enrollment and randomization.

## Data Availability

Not applicable.

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
