# Peer review of "Melatonin versus Sleep Deprivation for Sleep Induction in Nap Electroencephalography: Protocol for a Prospective Randomized Crossover Trial in Children and Young Adults with Epilepsy"

_metabolites, 2023, doi:10.3390/metabo13030383_

Round 1
Reviewer 1 Report
This is an interesting study to examine the impact of sleep deprivation and melatonin on stable epilepsy. This was submitted as a protocol versus a complete study with results. The review is based solely on the protocol.
Some criteria need better definition:
1. 50% of physiological sleep is still quite variable. What additional guidelines or info can be provided? Can patients wear an m-health device (like a watch with a sleep sensor) to better quantify heart rate and sleep the night before the EEG versus depending no caretaker observation? What ranges of sleep deprivation are considered for each age group and what are other mitigating factors and how will they be considered? Will there be sleep logs are other measures to identify what the patient's normal sleep habits and quality are to have more context for assessing the deprivation?
2. More information and references are needed for sample size calculation.
3. Randomization is good but given the vast age range and features, the investigators may need to consider full randomization could still result in unintentionally biased groups. An algorithm should be considered to mitigate these factors if possible.
4. The data analysis plan is insufficient. With EEG data, it should be fairly straightfoward to run some time series analysis and machine learning of the EEG to better assess differences that may not be picked up in standard statistical analysis of raw features. Additional unsupervised methods could also be helpful for assessing predictive features. The sample size proposed is too small for deep learning but cross-fold validation could provide some exploratory insights. Finally, the authors need to provide a specific list of features they wish to assess via quantitative statistical analysis.
Reviewer 2 Report
In the current manuscript, the authors proposed a Study Protocol on "Melatonin versus Sleep Deprivation for Sleep Induction in Nap Electroencephalography: Protocol for a Prospective Randomized Crossover Trial in Children and Young Adults with Epilepsy".
The authors suggest that this current study will allow a standardized protocol for sleep induction to be applied to еelectroencephalography (EEG) recordings in pediatric age. In the current presentation, the planned experiment looks like an interesting original study, however, it lacks some important components. The authors propose to compare the effectiveness of melatonin sleep induction (at a dose of 3 mg for patients < 15 kg; 5 mg for patients >15 kg) with deprivation of 50% of physiological sleep. In this regard, it is not entirely clear why a placebo is not supposed to be given along with melatonin. Since there are no plans to study a placebo group, it will not be possible to isolate the role of a mild hypnotic such as melatonin in inducing sleep. Also, based on the proposed experiments, the authors plan to develop a standardized protocol for sleep induction to be applied to EEG recordings in pediatric age. However, this will be quite difficult to do, since it is not planned to study a group where they would simultaneously give melatonin and perform deprivation of 50% of physiological sleep to induce sleep.
The rest of the planned study looks interesting. The current Study Protocol is clear enough to be published in the journal Metabolites after the authors justify why they would not use a placebo group in their study.
Reviewer 3 Report
In this STUDY PROTOCOL, the authors detailed their non-inferiority, randomized cross-over trial where 60 children and young adult patients with epilepsy will be randomized to receive oral solution of melatonin 3-5 mg (depending on the body weight of the participants) or partial sleep deprivation (more than 50% physiological sleep) as an intervention for sleep induction in nap EEG examinations. I have only a few comments for the study protocol.
If this study are properly carried out, it has the potential to offer a standardized protocol for sleep 39 induction to be applied to EEG recordings in pediatric age.. I only have a few comments to further improve the quality of the authors’ paper. I have outlined these issues below:
1.
Page 5 Sample size calculation
The calculated limit of non-inferiority of the mean difference between the two treatments was -3 minutes. Please specify “the mean difference of what”. sleep onset latency?
2.
Page 5 Melatonin and 6-hydroxymelatonin levels
When will Urinary concentration of melatonin and of its main metabolite 6-hydroxymelatonin be collected? Please specify it.
3.
Page 4
2.5. Primary outcome measures
What is the non-inferiority null hypothesis?
Please specify the definition that the clinical trial is successful when the study has more than one primary outcome measure.
Sleep onset latency, WASO and sleep duration all exceed the non-inferiority margin? Or sleep onset latency only? Or any one of them? Or one by one, in a specific order?
Will correction for multiple tests be applied?
4.
Page 4
2.4. Interventions
For participants who are signed to melatonin treatment, if the investigators know that the participant had insomnia the previous night before the EEG examinations (they might therefore have a nap in the morning of the day of EEG examinations), what would the investigators do to reduce possible confounding factors?
In the reviewer’s opinion, the above-mentioned issues need to be addressed by the authors.
